# Asynchronous stochastic gradient descent with decoupled backpropagation and layer-wise updates

## Abstract

The increasing size of deep learning models has created the need for more efficient alternatives to the standard error backpropagation algorithm, that make better use of asynchronous, parallel and distributed computing. One major shortcoming of backpropagation is the interlocking between the forward phase of the algorithm, which computes a global loss, and the backward phase where the loss is back-propagated through all layers to compute the gradients, which are used to update the network parameters. To address this problem, we propose a method that paral-lelises SGD updates across the layers of a model by asynchronously updating them from multiple threads. Furthermore, since we observe that the forward pass is of-ten much faster than the backward pass, we use separate threads for the forward and backward pass calculations, which allows us to use a higher ratio of forward to backward threads than the usual 1:1 ratio, reducing the overall staleness of the parameters. Thus, our approach performs asynchronous stochastic gradient de-scent using separate threads for the loss (forward) and gradient (backward) com-putations and performs layer-wise partial updates to parameters in a distributed way. We show that this approach yields close to state-of-the-art results while running up to $2.97\times$ faster than Hogwild! scaled on multiple devices (Locally-Partitioned-Asynchronous-Parallel SGD). We theoretically prove the convergence of the algorithm using a novel theoretical framework based on stochastic differ-ential equations and the drift diffusion process, by modeling the asynchronous parameter updates as a stochastic process.

## 1 Introduction

Scaling up modern deep learning models requires massive resources and training time. Asyn-chronous parallel and distributed methods for training them using backpropagation play a very important role in easing the demanding resource requirements for training these models. Back-propagation (BP) (Werbos, 1982) has established itself as the de facto standard method for learning in deep neural networks (DNN). Although BP achieves state-of-the-art accuracy on literally all rel-evant machine learning tasks, it comes with a number of inconvenient properties that prohibit an efficient implementation at scale.

BP is a two-phase synchronous learning strategy in which the first phase (forward pass) computes the training loss, $\mathcal{L}$, given the current network parameters and a batch of data. In the second phase, the gradients are propagated backwards through the network to determine each parameter's contribution to the error, using the same weights (transposed) as in the forward pass (see Equation 1). BP suffers from update locking, where a layer can only be updated after the previous layer has been updated. Furthermore, the computation of gradients can only be started after the loss has been calculated in the forward pass.

Moreover, the backward pass usually requires approximately twice as long as the forward pass (Ku-mar et al., 2021). The bulk of the computational load comes from the number of matrix multiplica-tions required during each phase. If we consider a DNN with $M$ layers, then at any layer $m \leq M$ with pre-activations $z_m = \theta_m y_{m-1}$ and post-activations $y_m = f(z_{m-1})$, the computations dur-ing the forward pass are dominated by one matrix multiplication $\theta_m y_{m-1}$. During the backward

pass, the computations at layer $m$ are dominated by two matrix multiplications:

$$\frac{\partial \mathcal{L}}{\partial \boldsymbol{\theta}_m} = f^{'}(\boldsymbol{\theta}_m \boldsymbol{y_{m-1}}) \times \boldsymbol{y_{m-1}}^{\top} \quad \text{and} \quad \frac{\partial \mathcal{L}}{\partial \boldsymbol{y_{m-1}}} = f^{'}(\boldsymbol{\theta}_m \boldsymbol{y_{m-1}}) \times \boldsymbol{\theta}_m^{\top}, \tag{1}$$

approximately doubling the compute budget required for the backward pass compared to the forward pass. In Eq. 1, $\boldsymbol{\theta}_m$ denotes the network weights at layer $m$ and $f'$ the partial derivative with respect to $\boldsymbol{\theta}_m$. This imbalance between forward and backward phase further complicates an efficient parallelization of BP, particularly in heterogeneous settings.

In this work, we propose a new approach to parallelize training of deep networks on non-convex objective functions by asynchronously performing the forward and backward passes at a layer-wise granularity in multiple separate threads that make lock-free updates to the parameters in shared memory. The lock-free updates address the locking problem, performing layer-wise updates mitigates the issue of conflicts between parameter updates, and performing the backward pass and updates using more threads than the forward pass mitigates the staleness problem. Specifically, the imbalance in execution time between forward and backward passes is taken care of by having twice as many backward threads than forward threads, breaking the 1:1 ratio of vanilla Backpropagation, therefore significantly speeding-up the training process.

In summary, the contributions of this paper are as follows:

1. We introduce a novel asynchronous formulation of Backpropagation which allows the forward pass and the backward pass to be executed separately and in parallel, which allows us to run more backward than forward threads. This approach accounts for the unequal time required by the forward and backward passes.

2. We propose to asynchronously update the model's parameters at a layer-wise granularity without using a locking mechanism which reduces staleness.

3. We give convergence guarantees of the algorithm to a stationary distribution centered around the local optima of conventional BP.

4. We show that the algorithm can reach state-of-the-art performances while being significantly faster than competing asynchronous algorithms.

## 2 RELATED WORK

**Asynchronous stochastic gradient descent (SGD).** Asynchronous SGD has a long history, starting from Baudet (1978); Bertsekas & Tsitsiklis (2015). Hogwild! (Recht et al., 2011) allows multiple processes to perform SGD without any locking mechanism on shared memory. Kungurtsev et al. (2021) proposed PASSM and PASSM+, where they partition the model parameters across the workers on the same device to perform SGD on the partitions. Chatterjee et al. (2022) decentralizes Hogwild! and PASSM+ to allow parameters or their partitions to be located on multiple devices and perform Local SGD on them. Zheng et al. (2017) compensate the delayed gradients with a gradient approximation at the current parameters. Unlike these methods, we run multiple backward passes in parallel on different devices and don't need any gradient compensation scheme.

Nadiradze et al. (2021) provides a theoretical framework to derive convergence guarantees for a wide variety of distributed methods. Mishchenko et al. (2022) proposes a method of "virtual iterates" to provide convergence guarantees independent of delays. More recently, Even et al. (2024) proposed a unified framework for convergence analysis of distributed algorithms based on the AGRAF framework. There have been lots of other analysis methods proposed for deriving convergence guarantees for asynchronous distributed SGD (see Assran et al. (2020) for a survey). In our work, we propose an entirely novel framework based on stochastic differential equations, and provide convergence guarantees of the algorithm to a stationary distribution centered around the local optima of conventional BP.

**Communication-efficient algorithms.** One of the bottlenecks when training on multiple devices or nodes in parallel is the synchronization step. The bigger or deeper the models get, the more time is consumed by synchronization. PowerSGD computes low-rank approximations of the gradients using power iteration methods. Poseidon (Zhang et al., 2017) also factorizes gradient matrices but interleaves their communication with the backward pass. Wen et al. (2017) and Alistarh et al.

(2017) quantize gradients to make them lightweight for communication. Like Zhang et al. (2017), we interleave the backward pass with gradients communication but without gradients averaging.

**Block local learning.** Dividing the network across multiple devices and performing local updates is a widely recognized approach in distributed learning. The backward passes of the different blocks can be done simultaneously. The global loss is used to provide feedback only to the output block while the remaining blocks get learning signals from auxiliary networks which each compute local targets. Jaderberg et al. (2016) models synthetic gradients through auxiliary networks. Ma et al. (2024) uses a shallower version of the network itself as the auxiliary network at each layer. Gomez et al. (2022) allows gradients to flow to k-neighboring blocks. Nøkland & Eidnes (2019) don't allow gradients to flow to neighboring blocks, and instead use an auxiliary matching loss and a local cross-entropy loss to compute the local error. Decoupled Parallel Backpropagation (Huo et al., 2018) does full Backpropagation but uses stored stale gradients in the blocks to avoid update locking, therefore needing additional memory buffers. Kappel et al. (2023) take a probabilistic approach by interpreting layers outputs as parameters of a probability distribution. Auxiliary networks provide local targets, which are used to train each block individually. Similar to these distributed paradigms, we mimic the execution of multiple backward passes in parallel by reordering the training sequence but without splitting the network explicitly during forward and backward propagation across devices and needing external buffers or architectural complexities.

# 3 METHODS

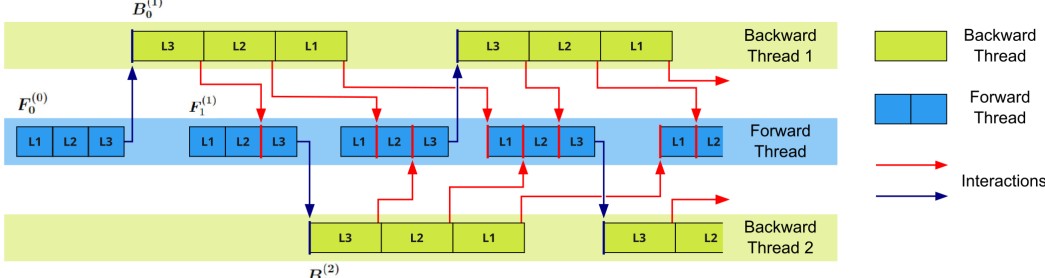

Figure 1: Illustration of decoupled backpropagation with separate threads for the forward and backward passes and the layer-wise updates. For each thread, the order of computations for a sample network with three layers denoted L1, L2 and L3, are shown. Arrows denote dependencies across threads. Within each thread, the computations for each layer are performed sequentially, whereas across threads, the dependencies are layer-wise. Interactions for 2 backward threads and a single forward thread are shown. This asynchronous interaction, along with layer-wise updates, reduces the staleness of parameters.

## 3.1 ASYNCHRONOUS FORMULATION OF BACKPROPAGATION

We introduce a new asynchronous stochastic gradient descent method where, instead of performing the forward and backward phases sequentially, we execute them in parallel and perform layer-wise parameter updates as soon as the gradients for a given layer are available. The dependencies between forward and backward phases are illustrated in Figure 1.

Since the gradient computation in the backward pass tends to consume more time than the loss calculation in the forward pass, we decouple these two into separate threads and use one forward thread and two backward threads to counterbalance the disproportionate execution time.

Figure 1 illustrates the interaction among threads based on one example. Initially, only the first forward pass, $F_0^{(0)}$, is performed. The resulting loss is then used in the first backward pass $B_0^{(1)}$, which starts in parallel to the second forward pass $F_1^{(1)}$. Once $F_1^{(1)}$ ends, its loss is used by $B_1^{(2)}$ running in parallel to the next forward pass and $B_0^{(1)}$.

## 3.2 LAYER-WISE UPDATES

Parallelizing the forward and backward passes can speed up training, but it violates several key assumptions of Backpropagation leading to sub-optimal convergence observed in different studies (Keuper & Preundt, 2016; Zheng et al., 2017). This happens because the losses and gradients are often calculated using inconsistent and outdated parameters.

To alleviate this problem, we update the layers as soon as the corresponding gradients are available from the backward pass. $F_1^{(1)}$ receives partial parameter updates from $B_0^{(1)}$ as soon as they are available. Therefore, the parameters used in $F_1^{(1)}$ will differ from those used in $F_0^{(0)}$ because some layers of the model would have been already updated by the thread $B_0^{(1)}$. On average, we can expect that the second half of the layers use a new set of parameters. It is important to note that the updates happen without any locking mechanism and asynchronous to the backward pass as done by Zhang et al. (2017).

## 3.3 SPEED-UP ANALYSIS

Before discussing experimental results, we study the potential speed-up of the asynchronous with layer-wise updates formulation over standard Backpropagation. To arrive at this result, we make the following assumptions to estimate the performance gain

- We assume that there are no delays between the end of a forward pass, the beginning of its corresponding backward pass and the next forward pass. This implies for example that as soon as $F_0^{(0)}$ ends, $F_1^{(1)}$ and $B_0^{(1)}$ begin immediately. Multiples backward threads are therefore running in parallel.
- Vanilla Backpropagation performs $b$ forward passes. We assume that this number also corresponds to the number of backward passes and the number of batches of data to be trained on.
- A forward pass lasts $T$ units of time and a backward pass $\beta T$ units of time, with a scaling factor $\beta > 1$ (expected to be at around 2 as show in appendix A.3).

The speed-up factor $\lambda$ observed can be express as the fraction between the estimated time taken by the standard BP, $T_1$, over the Async version of BP, $T_2$. Due to the sequential nature of BP, $T_1 = (1 + \beta)bT$. Similarly, $T_2 = (b + \beta)T$ since the backward pass runs parallel to the forward pass. The speed-up factor $\lambda$ is given by:

$$\lambda = \frac{(1 + \beta)b}{(b + \beta)} \ .$$

Considering a large number of batches, $b \to \infty$, we have

$$\lambda = 1 + \beta \ .$$

Hence, the maximum achievable speedup is expected to be $1 + \beta$, where $\beta$ is the scaling factor of the backward pass time. In practice, the speed-up factor $\lambda$ can be influenced by multiples factors like data loading which is sometimes a bottleneck (Leclerc et al., 2023; Isenko et al., 2022), or the system overhead, which reduce the achievable speedup.

## 3.4 STALENESS ANALYSIS

Here, we demonstrate the advantage of applying layer-wise updates (LU) compared to block updates (BU). BU refers to performing updates only after the entire backward pass is complete, a technique used in various previous asynchronous learning algorithms, e.g. (Recht et al., 2011; Chatterjee et al., 2022; Zheng et al., 2017). We use the same notation as in section 3.3.

To express this formally, we define the relative staleness $\tau$ of BU compared to LU as the time delay between when the gradients become available and when they are used to update the model weights. The intuition behind this lies in the fact that the more the updates are postponed, the more likely the gradients will become stale. The staleness will only increase with time and accumulate across the

layers. Assuming that the time required to compute the gradients for each layer is uniform and equal to $\frac{\beta T}{M}$, the relative staleness is expressed as $\tau = \frac{\beta T(M-1)}{2}$.

To see this, we use that by definition, the staleness increases as we approach the output layer. At any layer $m$, the layer-wise staleness $\tau_m = \frac{\beta T}{M}m$. Averaging over the layers, we have

$$\tau \;=\; \sum_{m=1}^{M} \tau_m \;=\; \frac{\beta T}{M} \sum_{m=1}^{M} m \;=\; \beta T \frac{(M-1)}{2}$$

Clearly, $\tau$ increases with the network's depth and the time required to perform one backward pass. Thus, the speedup is expected to scale approximately linearly with the network depth, showing the advantage of LU over BU for large $M$.

### 3.5 ALGORITHM

The Async BP algorithm is illustrated in Figure 1 and described in Algorithm listing 1. The algorithm consists of two components: a single forward thread and multiple backward threads. All threads work independently and asynchronously without any locking mechanism.

The forward thread is solely responsible for computing the loss $\mathcal{L}_i(\boldsymbol{\theta}^u, \boldsymbol{x}_i, \boldsymbol{y}_i)$, given the current mini batch of data $(\boldsymbol{x}_i, \boldsymbol{y}_i) \in \mathcal{D}$ and the latest set of updated weights $\boldsymbol{\theta}^u$. Since the algorithm works asynchronously, the weights $\boldsymbol{\theta}^u$ can be updated by any backward thread even while forward pass progresses. Once the forward pass is done, $\mathcal{L}_i$ is sent to one of the backward threads and the forward thread moves to the next batch of data.

In parallel, a backward thread $k$ receives a loss $\mathcal{L}_j$ and performs the backward pass. At each layer $m$, the gradients $G(\boldsymbol{\theta}^v_{m,k}) = \frac{\partial \mathcal{L}_j}{\partial \boldsymbol{\theta}^v_{m,k}}$ are computed, after which $\boldsymbol{\theta}^v_{m,k}$ is immediately used to update the forward thread parameters. Note that the backward thread here can potentially calculate the gradients for different values of parameters $\boldsymbol{\theta}^v_{m,k}$ than the ones used for the forward pass $\boldsymbol{\theta}^u$. In Section 5 and appendix B we show that this algorithm closely approximates conventional stochastic gradient descent, if asynchronous parameter updates arrive sufficiently frequently.

---

**Algorithm 1 Async BP with decoupled partial updates**

---

**Forward thread**

---

    **Given:** Data: $(x_i, y_i) \in \mathcal{D}$, latest up-to-date parameters: $\theta^u$

    Compute $\mathcal{L}_i(\theta^u, x_i, y_i)$

    send($\mathcal{L}_i$)          // send loss to a backward thread

---

**Backward Thread $k$**    **(running in parallel to the forward thread)**

---

    **Given:** Loss: $\mathcal{L}_j$, learning rate: $\eta$

    **for** layer $m \in [\text{M},1]$ **do**

        Compute $G(\theta^v_{m,k})$

        $\theta^{v+1}_{m,k} \leftarrow \theta^v_{m,k} - \eta.G$

        $\theta_m \leftarrow \theta^{v+1}_{m,k}$        // asynchronously update forward thread

    **end for**

---

## 4 RESULTS

We evaluate our method on three vision tasks, CIFAR-10, CIFAR-100 and Imagenet, and on one sequence modeling task: IMDb sentiment analysis. We use Resnet18 and Resnet50 architectures for vision tasks and a LSTM network for sequence modelling. These networks are trained on a machine with 3 NVIDIA A100 80GB PCIe GPUs with two AMD EPYC CPUs sockets of 64 cores each. The experiment code is based on the C++ frontend of Torch (Paszke et al., 2019) - Libtorch.

The Performance of these tasks is compared to Locally-Asynchronous-Parallel SGD (LAPSGD) and Locally-Partitioned-Asynchronous-Parallel SGD (LPPSGD) (Chatterjee et al., 2022). These methods extend the well-known Hogwild! algorithm and Partitioned Asynchronous Stochastic Subgradient (PASSM+) to multiple devices, respectively.

We record the achieved accuracy on the tasks and the wall-clock time to reach a target accuracy (TTA). If not stated otherwise, this accuracy is chosen to be the best accuracy achieved by the worst performing algorithm. We used the code made available by Chatterjee et al. (2022) which uses Pytorch Distributed Data-Parallel API.

### 4.1 Asynchnous training of vision tasks

We follow the training protocol of LAPSGD and LPPSGD and chose the number of processes per GPU to be 1 since the GPU utilization was close to 100%. We trained them with a batch size of 128 per-rank. We used Stochastic gradient descent (SGD) for both Async BU and LU, with an initial learning rate of 0.005 for 5 epochs and 0.015 after the warm-up phase, a momentum of 0.9 and a weight decay of $5 \times 110^{-2}$. We use a cosine annealing schedule with a $T_{max}$ of 110. We trained Resnet-50 on Imagenet-1K task (Table 5) for a total of 250 epochs with the same learning rate but with a cosine schedule with $T_{max}$ of 250 and weight decay of $3.5 \times 110^{-2}$. Although, the simulations were run with cosine annealing scheduler, implying the ideal number of training epochs, early stopping was applied, i.e. training was stopped if no improvement of the accuracy was achieved for 30 epochs.

As shown below, Async LU achieves the highest accuracies while Async BU converges the fastest in terms of time to reach the target accuracy. The CIFAR-10 and CIFAR-100 results are presented in Tables 1, 2 and Tables 3, 4 respectively. In Tables 1 and 3, the time to target accuracy (TTA) is chosen to be the time taken to achieved the best accuracy reached by the worst algorithm. Whereas in Tables 2 and 4, it represents taken by an algorithm achieve its best accuracy. Async BU achieves a speed-up of up to $2.97\times$ over LPPSGD on CIFAR100 (see Table 3). The poor performance of both LAPSGD and LPPSGD can be explained by the influence of staleness, thus requiring large number of training epochs.

We also achieved promising results on the ImageNet-1k dataset (see Table 5). Async LU achieved 73% accuracy $\times 3$ faster than Backpropagation on single GPU, showing potential of ideal linear scaling. An extensive comparison of Async LU with multi-GPU Backpropagation (Data Distributed Parallel) is provided in appendix A.2.

Although Async BU converges quicker than Async LU, it reaches lower accuracy. This is particularly visible on CIFAR100, a harder task than CIFAR10 (see Figures 2 and 3). Overall, Async BU showed a good balance between convergence speed and reduction of staleness.

Table 1: Comparison of Async LU, Async BU, LAPSGD and LPPSGD based on time to reach accuracy (TTA): 87% for ResNet18 and 89% for ResNet50, and the number of epochs to reach the target for 3 runs on CIFAR10.

| Network architecture | Training method | TTA (in seconds) mean ± std | Epochs mean ± std |
|---|---|---|---|
| ResNet-18 | Async LU | 223.8 ± 28 | 65 ± 10 |
| | Async BU | 173.4 ± 2 | 64 ± 2 |
| | LAPSGD | 706.3 ± 13 | 104 ± 2 |
| | LPPSGD | 461.0 ± 7 | 86 ± 1 |
| ResNet-50 | Async LU | 737.9 ± 24 | 86 ± 6 |
| | Async BU | 700.6 ± 20 | 86 ± 1 |
| | LAPSGD | 999.5 ± 38 | 117 ± 3 |
| | LPPSGD | 863.4 ± 35 | 119 ± 1 |

Table 2: Comparison of Async LU, Async BU, LAPSGD, and LPPSGD based on best accuracy, time to reach accuracy (TTA), and epoch at the accuracy is achieved for 3 runs on CIFAR10.

| Network architecture | Training method | Best accuracy mean ± std | TTA (in seconds) mean ± std | Epochs mean ± std |
|---|---|---|---|---|
| ResNet-18 | SGD | $93.7 \pm 0.16$ | $\pm$ | $114 \pm 1$ |
| | Async LU | $93.7 \pm 0.28$ | $380.7 \pm 33$ | $114 \pm 9$ |
| | Async BU | $92.7 \pm 0.16$ | $308.5 \pm 14$ | $115 \pm 2$ |
| | LAPSGD | $88.2 \pm 0.43$ | $799.5 \pm 20$ | $118 \pm 2$ |
| | LPPSGD | $87.8 \pm 0.09$ | $523.4 \pm 15$ | $97 \pm 1$ |
| ResNet-50 | SGD | $94.1 \pm 0.20$ | $\pm$ | $99 \pm 5$ |
| | Async LU | $93.9 \pm 0.10$ | $1038.8 \pm 61$ | $121 \pm 6$ |
| | Async BU | $93.2 \pm 0.25$ | $953.7 \pm 73$ | $116 \pm 8$ |
| | LAPSGD | $89.7 \pm 0.27$ | $1098.9 \pm 30$ | $117 \pm 3$ |
| | LPPSGD | $89.3 \pm 0.43$ | $888.4 \pm 37$ | $119 \pm 1$ |

Table 3: Comparison of Async LU, Async BU, LAPSGD and LPPSGD based on time to reach accuracy (TTA): 60% for ResNet18 and 63% for ResNet50, and the number of epochs to reach the target for 3 runs on CIFAR100.

| Network architecture | Training method | TTA (in seconds) mean ± std | Epochs mean ± std |
|---|---|---|---|
| ResNet-18 | Async LU | $226.4 \pm 10$ | $69 \pm 7$ |
| | Async BU | $155.4 \pm 10$ | $57 \pm 4$ |
| | LAPSGD | $667.1 \pm 8$ | $99 \pm 2$ |
| | LPPSGD | $672.9 \pm 11$ | $100 \pm 2$ |
| ResNet-50 | Async LU | $622.7 \pm 32$ | $70 \pm 5$ |
| | Async BU | $641.8 \pm 16$ | $81 \pm 3$ |
| | LAPSGD | $1006.6 \pm 13$ | $107 \pm 1$ |
| | LPPSGD | $1016.0 \pm 6$ | $107 \pm 1$ |

## 4.2 ASYNCHRONOUS TRAINING OF SEQUENCE MODELLING TASK

For demonstrating Async BP training on sequence modelling, we evaluated an LSTM networks on the IMDb dataset (Maas et al., 2011). Sentiment analysis is the task of classifying the polarity of a given text. We used a 2-Layer LSTM network with 256 hidden dimensions to evaluate this task. We trained the network until convergence using the Adam optimizer with an initial learning rate of $1 \times 10^{-2}$. Results are shown in Table 6.

We observe that although both Async LU and Async BU achieve the same accuracy, performing layer-wise updates reduces the training time and number of steps to convergence almost by half (see Figure 4 in Appendix A) highlighting again the importance of this strategy.

## 5 THEORETICAL ANALYSIS OF CONVERGENCE

Here, we theoretically analyse the convergence behavior of the algorithm outlined above. For the theoretical analysis, we consider the general case of multiple threads, acting on the parameter set $\boldsymbol{\theta}$, such that the threads interact asynchronously and can work on outdated versions of the parameters. We model the evolution of the learning algorithm as a continuous-time stochastic process (Bellec et al., 2017) to simplify the analysis. This assumption is justified by the fact that learning rates are typically small, and therefore the evolution of network parameters is nearly continuous.

In the model studied here, the stochastic interaction between threads is modelled as noise induced by random interference of network parameters. To arrive at this model, we use the fact that the dynamics of conventional stochastic gradient descent (SGD) can be modelled as the system of stochastic

Table 4: Comparison of Async LU, Async BU, LAPSGD and LPPSGD based on best accuracy, time to reach accuracy (TTA), and epoch at the accuracy is achieved for 3 runs on CIFAR100.

| Network architecture | Training method | Best accuracy mean ± std | TTA (in seconds) mean ± std | Epochs mean ± std |
|---|---|---|---|---|
| ResNet-18 | SGD | $73.7 \pm 0.08$ | $\pm$ | $99 \pm 4$ |
| | Async LU | $73.9 \pm 0.38$ | $339.0 \pm 10$ | $114 \pm 9$ |
| | Async BU | $71.8 \pm 0.08$ | $277.2 \pm 17$ | $103 \pm 5$ |
| | LAPSGD | $61.3 \pm 0.23$ | $762.5 \pm 15$ | $114 \pm 3$ |
| | LPPSGD | $61.0 \pm 0.17$ | $742.0 \pm 19$ | $110 \pm 2$ |
| ResNet-50 | SGD | $76.8 \pm 0.44$ | $\pm$ | $104 \pm 1$ |
| | Async LU | $76.2 \pm 0.57$ | $1071.0 \pm 80$ | $122 \pm 6$ |
| | Async BU | $73.7 \pm 0.23$ | $956.2 \pm 28$ | $123 \pm 4$ |
| | LAPSGD | $63.7 \pm 0.33$ | $1110.1 \pm 27$ | $118 \pm 3$ |
| | LPPSGD | $63.5 \pm 0.10$ | $1122.1 \pm 24$ | $119 \pm 2$ |

Table 5: Classification accuracy (% correct) for Async LU (3 GPUs) and vanilla backpropagation (single GPU) on the ImageNet task.

| Network architecture | training method | test-accuracy | train-accuracy | TTA (in 1000 seconds) |
|---|---|---|---|---|
| ResNet-50 | Async LU | 73.42 | 93.08 | 134.24 |
| | BP (single GPU) | 73.40 | 91.90 | 403.77 |

differential equations that determine the dynamics of the parameter vector $\boldsymbol{\theta}$

$$d\theta_k = -\eta \frac{\partial}{\partial \theta_k} \mathcal{L}(\boldsymbol{\theta}) dt \; + \; \frac{\eta \, \sigma_{\text{SGD}}}{\sqrt{2}} \, d\mathcal{W}_k \; , \tag{2}$$

with learning rate $\eta$ and where $d\mathcal{W}_k$ are stochastic changes of the Wiener processes.

Eq. 2 describes the dynamics of a single parameter $\theta_k$. The dynamics is determined by the gradient of the loss function $\mathcal{L}$, and the noise induced by using small mini-batches modelled here as idealized Wiener process with amplitude $\sigma_{\text{SGD}}$. Because of this noise, SGD does not strictly converge to a local optimum but maintains a stationary distribution $p^*(\boldsymbol{\theta}_k) \propto e^{-\frac{1}{\eta}\mathcal{L}(\boldsymbol{\theta}_k)}$, that assigns most of the probability mass to parameter vectors that reside close to local optima (Bellec et al., 2017).

In the concurrent variant of SGD studied here, however, the dynamics is determined by perturbed gradients for different stale parameters. When updating the network using the described asynchronous approach without locking, we potentially introduce noise in the form of partially stale parameters or from one thread overwriting the updates of another. This noise will introduce a deviation from the ideal parameter vector $\boldsymbol{\theta}$. We model this deviation as additive Gaussian noise $\boldsymbol{\xi} \sim \mathcal{N}(0, \sigma_{\text{STALE}})$ to the current parameter vector with variance $\sigma_{\text{STALE}}$. To approximate the noisy loss function, we use a first-order Taylor expansion around the noise-free parameters:

$$\begin{aligned} \mathcal{L}(\boldsymbol{\theta} + \boldsymbol{\xi}) &= \mathcal{L}(\boldsymbol{\theta}) + \nabla_\theta \mathcal{L}(\boldsymbol{\theta})^\top \boldsymbol{\xi} + \mathcal{O}(\sigma^2) \\ &\approx \mathcal{L}(\boldsymbol{\theta}) + \nabla_\theta \mathcal{L}(\boldsymbol{\theta})^\top \boldsymbol{\xi} \, , \end{aligned} \tag{3}$$

and thus the gradient can be approximated as

$$\nabla_\theta \mathcal{L}(\boldsymbol{\theta} + \boldsymbol{\xi}, \boldsymbol{X}, \boldsymbol{Y}) \; \approx \; \nabla_\theta \mathcal{L}(\boldsymbol{\theta}) + \nabla_\theta^2 \mathcal{L}(\boldsymbol{\theta})^\top \boldsymbol{\xi} \, . \tag{4}$$

Based on this, we can express the update rule as a Stochastic Differential Equation (SDE) and model the various noise terms using a Wiener Process $\mathcal{W}$. The noise sources in the learning dynamics come from two main sources, (1) noise caused by stochastic gradient descent, and (2) noise caused by learning with outdated parameters. We model the former as additive noise with amplitude $\sigma_{\text{STALE}}$ and the latter using the Taylor approximation Eq. (4). Using this, we can write the approximate dynamics of the parameter vector $\boldsymbol{\theta}$ as the stochastic differential equation

$$d\theta_k = \mu_k(\boldsymbol{\theta}, t) + \sqrt{D_k(\boldsymbol{\theta})} \, d\mathcal{W}_k \; , \tag{5}$$

Table 6: Comparison of Async LU, Async BU based on best accuracy, time to reach accuracy (TTA), and epoch at the accuracy is achieved for 3 runs on IMDb

| Network architecture | | best accuracy mean±std | TTA(in seconds) mean±std | epoch mean±std |
|---|---|---|---|---|
| LSTM | Async LU | 85.15±0.15 | 49.3±9.46 | 6±1 |
| | Async BU | 85.06±0.59 | 83.41±12.56 | 12±2 |

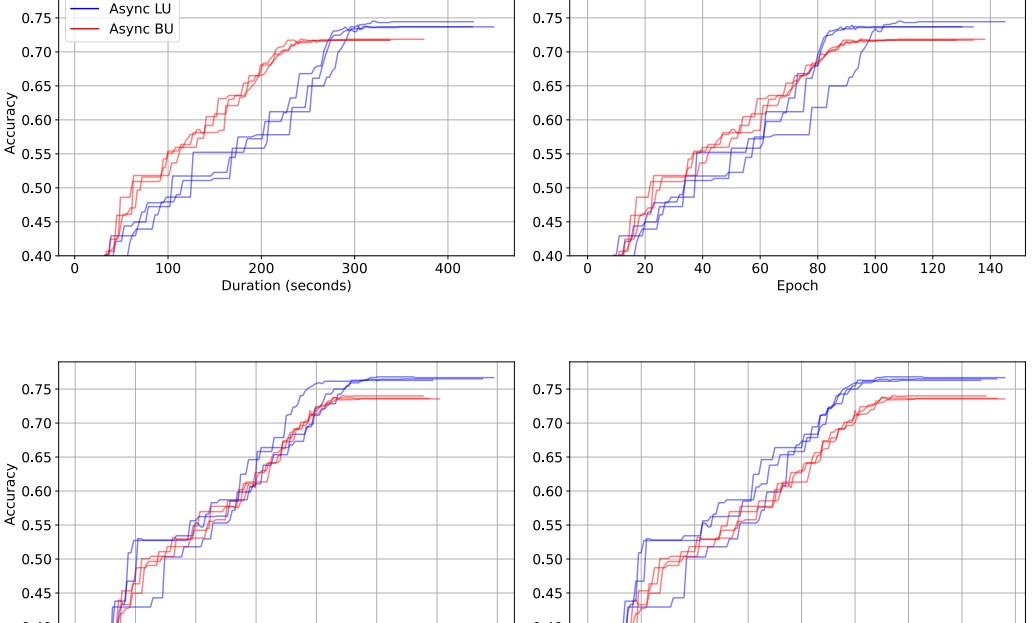

Figure 2: Learning curves of Asynchronous SGD with layer-wise updates (Async LU) and Block updates (Async BU) on the CIFAR100 dataset. 3 independent runs are shown for each class.

with

$$\mu_k(\boldsymbol{\theta}) = -\eta \frac{\partial}{\partial \theta_k} \mathcal{L}(\boldsymbol{\theta})$$

$$D_k(\boldsymbol{\theta}) = \frac{\eta^2 \sigma_{\text{SGD}}^2}{2} + \frac{\eta^2 \sigma_{\text{STALE}}^2}{2} \sum_l \frac{\partial^2}{\partial \theta_k \partial \theta_l} \mathcal{L}(\boldsymbol{\theta}) \,, \tag{6}$$

where $\mu_k$ is the drift and $D_k$ the diffusion of the SDE.

In Appendix B we study the stationary distribution of this parameter dynamics. We show that the stationary distribution is a close approximation to $p^*$ of SGD, which is perfectly recovered if $\sigma_{\text{STALE}}$ is small compared to $\sigma_{\text{SGD}}$, i.e. if the effect of staleness is small compared to the noise induced by minibatch sampling.

## 6 DISCUSSION

In this work, we introduced a novel asynchronous approach to train deep neural networks that decouples the forward and backward passes and performs layer-wise parameter updates. Our method addresses key limitations of standard backpropagation by allowing parallel execution of forward and backward passes and mitigating update locking through asynchronous layer-wise updates.

The experimental results demonstrate that our approach can achieve comparable or better accuracy than synchronous backpropagation and other asynchronous methods across multiple vision and lan-

guage tasks, while providing significant speedups in training time. On CIFAR-10 and CIFAR-100, we observed speedups of up to $2.97\times$ compared to asynchronous SGD covering a broad range of paradigms. The method also showed promising results on a sentiment analysis task and the ImageNet classification task where it reached close to ideal scaling.

Our theoretical analysis, based on modeling the learning dynamics as a continuous-time stochastic process, provides convergence guarantees and shows that the algorithm converges to a stationary distribution closely approximating that of standard SGD under certain conditions. This offers a solid foundation for understanding the behavior of our asynchronous approach.

While our implementation using C++ and LibTorch demonstrated the potential of this method, we also identified some limitations related to GPU resource allocation in SIMT architectures. Future work could explore optimizing the implementation for more efficient GPU utilization, or investigating hybrid CPU-GPU approaches to fully leverage the benefits of asynchronous execution.

Overall, this work presents a promising direction for scaling up deep learning through asynchronous, decoupled updates. The approach has the potential to enable more efficient training of large-scale models, particularly in distributed and heterogeneous computing environments. Further research could explore extensions to even larger models, additional tasks, and more diverse hardware setups to fully realize the potential of this asynchronous training paradigm.

## REPRODUCIBILITY

We ensure that the results presented in this paper are easily reproducible using just the information provided in the main text as well as the supplement. Details of the models used in our simulations are presented in the main paper and further elaborated in the supplement. We provide additional details and statistics over multiple runs in the supplement section A.4. We use publicly available libraries and datasets in our simulations. We will further provide the source code to the reviewers and ACs in an anonymous repository once the discussion forums are opened. This included code will also contain "readme" texts to facilitate easy reproducibility. The theoretical analysis provided in section 5 is derived in the supplement.

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

## A ADDITIONAL RESULTS

### A.1 LEARNING CURVES

Here, we provide additional details to the results provided in the main text. Figures 3 and 4 show the learning dynamics of Asynchrounous Backpropagation with blocks updates (Async BU) and with layer-wise updates (Async LU) on CIFAR10 and IMDb respectively. The difference in convergence speed and accuracy observed with CIFAR100 2 is less noticeable on CIFAR10, probably because it is a simpler task. However, we clearly see the advantage of Async LU on the IMDb, where it not only converges faster but also to similar accuracy.

Figures 5 and 6 compare the training curses of sequential SGD with Async LU respectively. We observed that Async LU needs more epochs to converge, ∼15 epochs more. When trained to a larger dataset (Figures 7 and 8), we can see that both Async LU and SGD seem to converge within the same number of epochs while Async LU scales almost linearly with the number of GPUs. We should take these results carefully given that accuracies are plots against the number of epochs and not the time since both are trained on different numbers of GPUs. Appendix A.2 gives a comparison with equal number of GPUs.

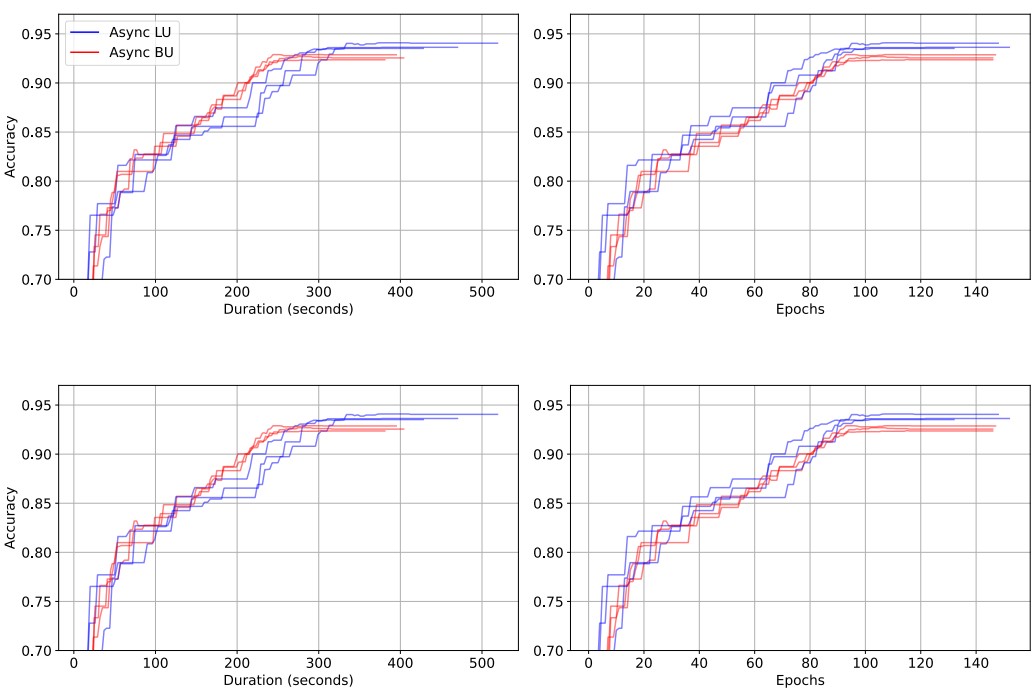

Figure 3: testing curves of Asynchronous SGD with layer-wise updates (Async LU) and Block updates (Async BU) for ResNet18 (top plots) and ResNet50 (bottom plots) on the CIFAR10 dataset.

### A.2 SPEED-UP COMPARISON WITH MULTI-GPU BACKPROPAGATION

Here we do a comparison of Asynchronous Backpropagation with layer-wise updates (Async LU) and multi-GPU Data Distributed Parallel(DDP) both trained on 3 GPUs residing on the same machine (described in section 4) to achieve their accuracies. Since Async LU uses only one forward pass, we set its batch size to be 128 and that of BP to 3× higher (384). Async LU was implemented on the c++ library of Pytorch, Libtorch, while multi-GPU SGD is implemented using Pytorch DataDistributedParallel (DDP) API. The hyperparameters used are the same as described in section 4.

To make the comparison fair across implementation platforms, the relative speed-up is calculated with respect to their single GPU implementation respectively.

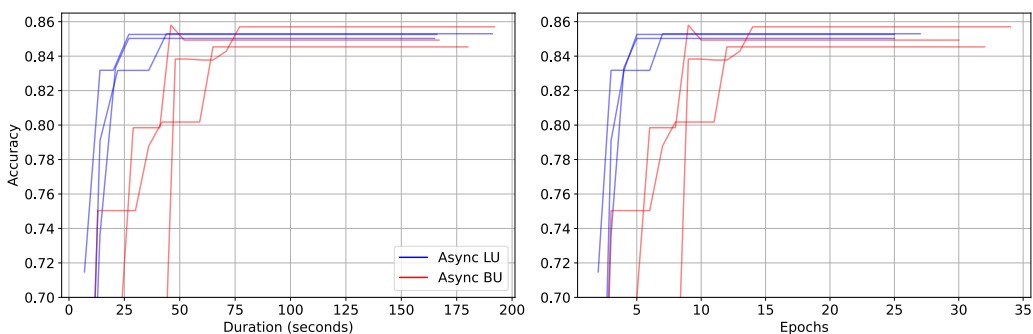

Figure 4: testing curves of Asynchronous SGD with layer-wise updates (Async LU) and Block updates (Async BU) on the IMDb dataset.

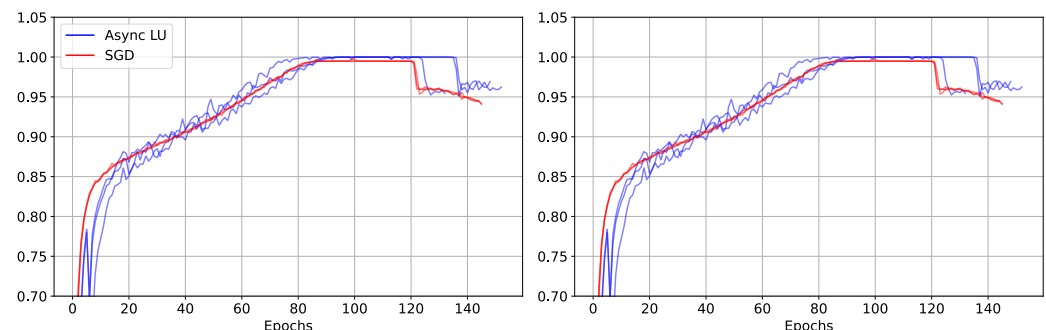

Figure 5: training curves of Asynchronous SGD with layer-wise updates (Async LU) on 3 GPUs and Single GPU SGD on CIFAR-10 dataset on ResNet18 (left) and ResNet50 (right).

In this Settings, DDP should clearly be at advantage since it uses a bigger batch size, all the GPUs are on the same machine and we optimized DDP loading process by persisting the data on the GPUs, reducing considerably the communication bottleneck, hence making the synchronization step faster. What we however observe is that Async LU achieves comparable relative speed-up over a single GPU compared to DDP on both CIFAR10 and CIFAR100. This shows the effectiveness of our asynchronous formulation (figure 1). We can expect Async LU to have greater advantage in a multi-node or heterogeneous setting because the synchronization barrier becomes problem.

Table 7: Comparison of Async LU and DDP both trained on 3GPUs based on their relative speed-up to single GPU implemention for 3 runs on CIFAR10

| Network architecture | Training method | Relative speed-up mean $\pm$ std |
|---|---|---|
| ResNet-18 | Async LU | $1.86 \pm 0.21$ |
| | DDP | $1.90 \pm 0.08$ |
| ResNet-50 | Async LU | $2.02 \pm 0.10$ |
| | DDP | $2.38 \pm 0.19$ |

## A.3 TIME MEASUREMENTS

Here we provide the results of a small-scale experiment on the timing measurement of forward and backward passes for CIFAR-100 with batch size 128 in table 9. As expected, a single backward pass requires $\sim 2\times$ than that of a single forward pass. Extensive experiments on this is provided by Kumar et al. (2021)

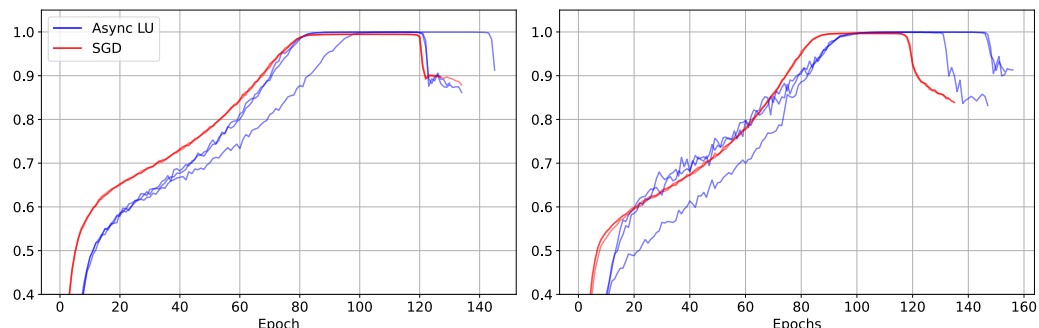

Figure 6: training curves of Asynchronous SGD with layer-wise updates (Async LU) on 3 GPUs and Single GPU SGD on the CIFAR-100 dataset on ResNet18 (left) and ResNet50 (right).

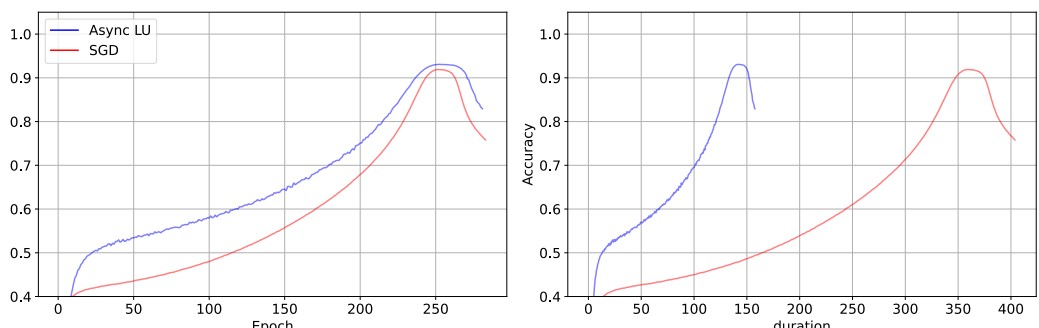

Figure 7: training curves of Asynchronous SGD with layer-wise updates (Async LU) on 3 GPUs and Single GPU SGD on the ImageNet dataset.

### A.4 Hyperparameters for the experiments

Hyperparameters used in training experiments presented in section 4 are documented in table 10

## B Convergence Proof

Here we provide the proof that the stochastic parameter dynamics, Eq. (5) of the main text, converges to a stationary distribution $p^*(\boldsymbol{\theta})$ given by

$$p^*(\boldsymbol{\theta}) = \frac{1}{\mathcal{Z}} \exp\left(\sum_k h_k(\boldsymbol{\theta})\right), \text{ with } h_k(\boldsymbol{\theta}) = \int \frac{\mu_k(\boldsymbol{\theta})}{D_k(\boldsymbol{\theta})}\, d\boldsymbol{\theta} - \ln |D_k(\boldsymbol{\theta})| + C . \qquad (7)$$

The proof is analogous to the derivation given in Bellec et al. (2017), and relies on stochastic calculus to determine the parameter dynamics in the infinite time limit. Since the dynamics include a noise term, the exact value of the parameters $\boldsymbol{\theta}(t)$ at a particular point in time $t > 0$ cannot be determined, but we can describe the distribution of parameters using the Fokker-Planck formalism, i.e. we describe the parameter distribution at time $t$ by a time-varying function $p_{\mathrm{FP}}(\boldsymbol{\theta}, t)$.

To arrive at an analytical solution for the stationary distribution, $p^*(\boldsymbol{\theta})$ we make the adiabatic assumption that noise in the parameters only has local effects, such that the diffusion due to noise in any parameter $\theta_j$ has negligible influence on dynamics in $\theta_k$, i.e. $\frac{\partial}{\partial \theta_j} D_k(\boldsymbol{\theta}) = 0, \forall j \neq k$. Using this assumption, it can be shown that, for the dynamics (6), $p_{\mathrm{FP}}(\boldsymbol{\theta}, t)$ converges to a unique stationary distribution in the limit of large $t$ and small noise $\sigma_{STALE}$. To prove the convergence to the stationary distribution, we show that it is kept invariant by the set of SDEs Eq. (6) and that it can be

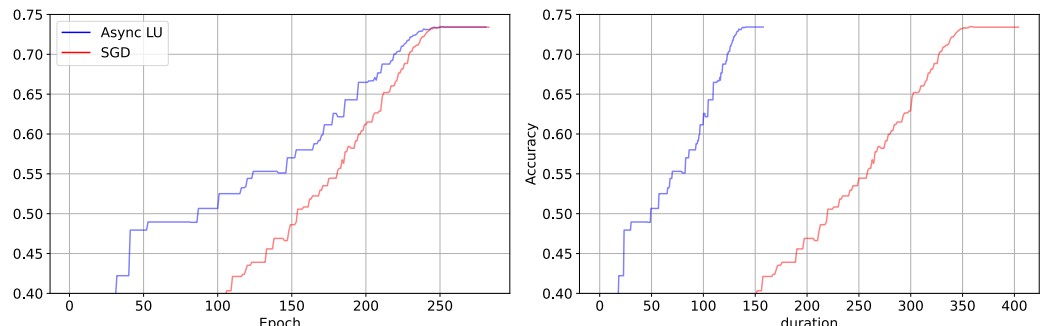

Figure 8: testing curves of Asynchronous SGD with layer-wise updates (Async LU) on 3 GPUs and Single GPU SGD on the ImageNet dataset.

Table 8: Comparison of Async LU and DDP both trained on 3GPUs based on their relative speed-up to single GPU implemention for 3 runs on CIFAR100

| Network architecture | Training method | Relative speed-up mean $\pm$ std |
|---|---|---|
| ResNet-18 | Async LU | $1.83 \pm 0.06$ |
|  | DDP | $1.86 \pm 0.21$ |
| ResNet-50 | Async LU | $2.02 \pm 0.13$ |
|  | DDP | $2.30 \pm 0.13$ |

reached from any initial condition. Eq. 6 implies a Fokker-Planck equation given by

$$\frac{\partial}{\partial t}p_{\mathrm{FP}}(\boldsymbol{\theta},t) = -\sum_k \frac{\partial}{\partial \theta_k}[\mu_k(\boldsymbol{\theta},t)p_{\mathrm{FP}}(\boldsymbol{\theta},t)] + \frac{\partial^2}{\partial \theta_k^2}[D_k(\boldsymbol{\theta},t)p_{\mathrm{FP}}(\boldsymbol{\theta},t)] \tag{8}$$

We show that, under the assumptions outlined above, the stochastic parameter dynamics Eq. (6) of the main text, converges to the stationary distribution $p^*(\boldsymbol{\theta})$ (Eq. (7)).

To arrive at this result, we plug in the assumed stationary distribution into Eq. (8) and show the equilibrium $\frac{\partial}{\partial t}p_{\mathrm{FP}}(\boldsymbol{\theta},t) = 0$, i.e.

$$
\begin{aligned}
\frac{\partial}{\partial t}p_{\mathrm{FP}}(\boldsymbol{\theta},t) \ =\ & -\sum_k \frac{\partial}{\partial \theta_k}[\mu_k(\boldsymbol{\theta})p_{\mathrm{FP}}(\boldsymbol{\theta},t)] \\
& + \frac{\partial^2}{\partial \theta_k^2}[D_k(\boldsymbol{\theta})p_{\mathrm{FP}}(\boldsymbol{\theta},t)] = 0 \\
\leftrightarrow\ & -\sum_k \frac{\partial}{\partial \theta_k}[\mu_k(\boldsymbol{\theta})p_{\mathrm{FP}}(\boldsymbol{\theta},t)] \\
& + \frac{\partial}{\partial \theta_k}\left[\left(\frac{\partial}{\partial \theta_k}D_k(\boldsymbol{\theta})\right)p_{\mathrm{FP}}(\boldsymbol{\theta},t)\right] \\
& + \frac{\partial}{\partial \theta_k}\left[D_k(\boldsymbol{\theta})\left(\frac{\partial}{\partial \theta_k}h_k(\boldsymbol{\theta})\right)p_{\mathrm{FP}}(\boldsymbol{\theta},t)\right],
\end{aligned}
\tag{9}
$$

Table 9: Timing measurement of forward and backward passes for CIFAR-100 with batch size 128. Averaged over all batches for 15 epochs.

| Network architecture | Forward pass (s) mean $\pm$ std | Backward pass (s) mean $\pm$ std |
|---|---|---|
| ResNet-18 | $0.0049 \pm$ 1E-04 | $0.0102 \pm$ 1E-04 |
| ResNet-50 | $0.0166 \pm$ 5E-05 | $0.0299 \pm$ 4E-05 |

Table 10

| | CIFAR-100 | | CIFAR-10 | | Imagenet | IMDb |
|---|---|---|---|---|---|---|
| hyperparameter | Resnet-18 | Resnet-50 | Resnet-18 | Resnet-50 | Resnet-50 | LSTM |
| batch_size | 128 | 128 | 128 | 128 | 250 | 75 |
| lr | 0.015 | 0.015 | 0.015 | 0.015 | 0.015 | 0.001 |
| momentum | 0.9 | 0.9 | 0.9 | 0.9 | 0.9 | 0.9 |
| T_max | 100 | 120 | 105 | 120 | 250 | 150 |
| warm_up_epochs | 5 | 5 | 3 | 5 | 5 | 0 |
| warm_up lr | 0.005 | 0.005 | 0.005 | 0.005 | 0.0035 | - |
| weight_decay | 0.005 | 0.005 | 0.005 | 0.005 | 0.0035 | 0 |

where we used the simplifying assumption, $\frac{\partial}{\partial \theta_j} D_k(\boldsymbol{\theta}) = 0, \forall j \neq k$, as outlined above. Next, using $\frac{\partial}{\partial \theta_j} h_k(\boldsymbol{\theta}) = \frac{1}{D_k(\boldsymbol{\theta},t)} \left( \mu_k(\boldsymbol{\theta},t) - \frac{\partial}{\partial \theta_j} D_k(\boldsymbol{\theta},t) \right)$, we get

$$
\begin{aligned}
\frac{\partial}{\partial t} p_{\text{FP}}(\boldsymbol{\theta},t) = 0 \quad \leftrightarrow \quad & -\sum_k \frac{\partial}{\partial \theta_k} [\mu_k(\boldsymbol{\theta},t) p_{\text{FP}}(\boldsymbol{\theta},t)] \\
& + \frac{\partial}{\partial \theta_k} \left[ \left( \frac{\partial}{\partial \theta_k} D_k(\boldsymbol{\theta},t) \right) p_{\text{FP}}(\boldsymbol{\theta},t) \right] \\
& + \frac{\partial}{\partial \theta_k} \left[ \left( \mu_k(\boldsymbol{\theta},t) - \frac{\partial}{\partial \theta_k} D_k(\boldsymbol{\theta},t) \right) p_{\text{FP}}(\boldsymbol{\theta},t) \right] = 0
\end{aligned}
\tag{10}
$$

This shows that the simplified dynamics, Eq. 6, leave the stationary distribution (7) unchanged.

This stationary distribution $p^*(\boldsymbol{\theta})$ is a close approximation to SGD. To see this, we study the maxima of the distribution, by taking the derivative

$$
\frac{\partial}{\partial \theta_k} h_k(\boldsymbol{\theta}) = \frac{\mu_k(\boldsymbol{\theta})}{D_k(\boldsymbol{\theta})} - \frac{\partial}{\partial \theta_k} \ln |D_k(\boldsymbol{\theta})| ,
\tag{11}
$$

which by inserting (6) can be written as

$$
\frac{\partial}{\partial \theta_k} h_k(\boldsymbol{\theta}) = -\frac{1}{\eta} \frac{\nabla_\theta \mathcal{L}(\boldsymbol{\theta}) + \sigma_{\text{STALE}}^2 \nabla_\theta^3 \mathcal{L}(\boldsymbol{\theta})}{\sigma_{\text{SGD}}^2 + \sigma_{\text{STALE}}^2 \nabla_\theta^2 \mathcal{L}(\boldsymbol{\theta})}
\tag{12}
$$

If $\sigma_{\text{STALE}}$ is small compared to $\sigma_{\text{SGD}}$ we recover the cannonical results for SGD $\frac{\partial}{\partial \theta_k} h_k(\boldsymbol{\theta}) \approx -\frac{1}{\eta} \frac{\nabla_\theta \mathcal{L}(\boldsymbol{\theta})}{\sigma_{\text{SGD}}^2}$, where smaller learning rates $\eta$ make the probability of reaching local optima more peaked. Distortion of local optima, which manifests in the second term in the nominator, only depend on third derivatives, which can be expected to be small for most neural network architectures with well-behaved non-linearities.

