# OpenReview forum: "Asynchronous stochastic gradient descent with decoupled backpropagation and layer-wise updates"
_ICLR.cc/2025/Conference — Submitted to ICLR 2025_

### Official Review · Reviewer_Mg9U · 2024-11-02

**Soundness:** 2
**Presentation:** 2
**Contribution:** 2
**Rating:** 5
**Confidence:** 4

**Summary:**

The paper is in the general area of parallelization of DNN training, at the intersection of Machine Learning and Systems. Specifically, the paper is motivated by an analysis of backpropagation from the point of view of parallelism. Specifically, the paper starts by observing that backprop through a DNNs is a heavily sequential process (since we have to propagate inputs layer-wise, and then back-propagate the other way). The paper then wonders what would happen if we were to decouple, e.g., the forward-backward processes, by executing them on different threads, in order to allow for strictly increased parallelism relative to the base process.

**Strengths:**

I found the paper’s motivation and its initial discussion, up to Section 3.2, sound, interesting, and worthwhile. If anything, I think the observation that the max speedup factor should be at most 3 (since in standard backprop we are executing 3 matrix multiplies in sequence, and in your loose version they could potentially happen completely in parallel, on average) could be stated earlier and arrived at less circuitously.

**Weaknesses:**

Weaknesses:

This point in the paper is unfortunately also pretty much where the quality of the submission starts to degrade. Specifically, I will list my concerns relative to the rest of the paper below, in sequence. (The authors may use this indexing in the discussion period.)

1. Section 4 jumps directly into the experimental results. Here, the authors seem to meet a few challenges in terms of the viability of their method, which they seem to obscure or completely ignore.

a. First is the fact that the sample complexity of their asynchronous algorithm (i.e. the number of data samples seen by SGD) is (probably, unless I misunderstand something, this is not discussed clearly) much higher than standard SGD with momentum: by this I mean that, due to asynchrony, their algorithm seems to need a lot more data passes to reach the same level of accuracy.
For instance, curiously, the authors choose 300 epochs for ImageNet training of ResNet50, which is of only 73% for their algorithm. By comparison, e.g. the FFCV recipe using synchronous SGD+momentum on a single A100 would reach the same level in 30 epochs and probably in lower wall-clock time. A standard accuracy for this network is 76.15 (SGD + momentum, 90 epochs), if memory serves me well.

The same curious behavior seems to occur for CIFAR10/100 experiments as well, where the accuracies presented significantly underperform what I seem to remember are the “right” values for standard training (e.g. around 93% on CIFAR10). Please see the FFCV examples or even Pytorch pretrained models for reasonably SOTA versions.

b. More broadly, I find it really quite curious that none of the tables presented contains an SGD baseline. It is great that the authors are able to outperform LAPSGD/LPPSGD–and frankly what they are doing does seem like a better idea for parallelization–but that doesn’t mean that one should ignore any other baseline. Please add baselines.

c. Generally, it is not so clear to me what are the broad implications of Section 4. OK, the results are better than LPP/LAP in terms of accuracy per time unit, but I thought that the general findings should be that the approach is better than “sequential” SGD?

2. The theoretical analysis of convergence is under a very non-standard model.
Specifically, the analysis seems to boil down to modeling SGD dynamics as a continuous process (which is fine), but then assuming that somehow the async SGD noise satisfies conditions which would allow it to be “linearized,” i.e. apply Taylor around the unperturbed parameters. It is really not clear to me why this is OK to do. Moreover, the authors don’t seem to be interested in providing a more standard analysis of SGD, e.g. along the lines of the Nadiraze et al. paper they seem to be well aware of.
Specifically, the analysis allows us to grok the rationale for why the scheme might work, but seems to be primarily motivated by expediency rather than technical precision.

**Questions:**

Please see issues above for major questions.

Some typos and minor observations:

- L249: “for *a* different values”

- 250: Appendix B should be capitalized

- 251: sufficiently frequently

- 275: the A100 GPU you are using is an unusual choice for this task: this might be helping your algorithm since it has large computational power, so the degree of asynchrony between threads may be reduced relative to a mid-range GPU.

- 292: Why did you choose 300 epochs for ImageNet? What happens with a lower (and more standard) number?

- Please consider moving figures closer to where they are referenced.

- Why no SGD baselines in tables?

- Your convergence curves in Figure 2 seem to suggest that in fact there is quite significant noise due to asynchrony (the staircase pattern). Can you show a similar curve for regular training? Can you maybe vary the degree of asynchrony?

Necessary Changes

1. Comprehensive baseline comparisons including standard SGD
2. Analysis of sample complexity increase and its practical implications, rather than sweeping it under the rug
3. Validation on more diverse hardware configurations, ablations on key components, detailed scalability analysis
4. Clearer theoretical justification or an alternative, more standard, analysis approach

---

> ### Author Response · Authors · 2024-11-21
>
> We thank the reviewer for feedback and suggestions for improvement. We respond to individual points below.
>
> * **Section 4 jumps directly into the experimental results. Here, the authors seem to meet a few challenges in terms of the viability of their method, which they seem to obscure or completely ignore.**:
> > Section 4 is dedicated to presenting the results of our experiments.  Before that, we provide a detailed description of the experimental setup, including the datasets, algorithms, and hardware utilized. Should any aspects require further clarification, we would be happy to provide additional details.
> * **First is the fact that the sample complexity of their asynchronous algorithm: by this, I mean that, due to asynchrony, their algorithm seems to need a lot more data passes to reach the same level of accuracy**:
> >You are right, our approach can be less sample-efficient than standard SGD. This well-known problem usually arises because of stale gradients, but it is not problematic since asynchronous methods have higher throughput to compensate for. Empirically, for CIFAR10 and ImageNet the same number of epochs is needed to reach the same performance as standard SGD but for CIFAR100, ~20 epochs more are needed. So our method is slightly worse for CIFAR100, but it is NOT much higher.
>
> * **For instance, curiously, the authors choose 300 epochs for ImageNet training of ResNet50, which is of only 73% for their algorithm. By comparison, e.g. the FFCV recipe using synchronous SGD+momentum on a single A100 would reach the same level in 30 epochs and probably in lower wall-clock time. A standard accuracy for this network is 76.15 (SGD + momentum, 90 epochs), if memory serves me well. The same curious behavior seems to occur for CIFAR10/100 experiments as well, where the accuracies presented significantly underperform what I seem to remember are the “right” values for standard training (e.g. around 93% on CIFAR10). Please see the FFCV examples or even Pytorch pretrained models for reasonably SOTA versions.**
> > You are right, there is a discrepancy between our accuracies and the one reported by Pytorch/FFCV.
> We used the Libtorch library for our experiments to benefit from C++ multithreading capacities since true multithreading doesn’t yet exist in Python. We have noticed an accuracy gap between Python and C++ accuracies, most probably because the model implementation and data preparation steps were written by us. On CIFAR10, our baseline is the best accuracy reached by SGD: ~93-94% and ~73% on ImageNet using Libtorch. We have added those values in the tables.
> Regarding FFCV, it is used to alleviate the data loading bottleneck through data preloading. It can be combined with our method, but we haven’t done that yet, because they don’t have a C++ API.
> * **More broadly, I find it really quite curious that none of the tables presented contains an SGD baseline. It is great that the authors are able to outperform LAPSGD/LPPSGD–and frankly what they are doing does seem like a better idea for parallelization–but that doesn’t mean that one should ignore any other baseline. Please add baselines.**:
> >Thank you for the remark.
> We have added accuracies for standard SGD (single GPU) in the table. However, we have a comparison with Distributed Data-Parallel (DDP), a multi-GPU implementation, in Appendix A2.
> * **Generally, it is not so clear to me what are the broad implications of Section 4. OK, the results are better than LPP/LAP in terms of accuracy per time unit, but I thought that the general findings should be that the approach is better than “sequential” SGD?**:
> >Our approach performs comparably to Distributed Data Parallel (DDP). However, it is important to note that the DDP implementation we use offers an advantage due to improvements in the data loading process. Specifically, by utilizing persistent workers, we were able to significantly speed up the data loading, which would otherwise be approximately twice as slow. In contrast, our implementation, which relies on the C++ API, does not benefit from this optimization. To the best of our knowledge, no existing asynchronous methods outperform SGD in this particular setup.
> * **The theoretical analysis of convergence is under a very non-standard model.**:
> >The first-order Taylor approximation provides only an estimate of how noise in general impacts the gradient of the loss with respect to the parameters, and consequently, how it affects the update step of SGD. We currently do not model the specific noise that arises from our asynchronous approach. We will look into providing a proof more along the lines of a “standard” approach as well in a future version of the paper.
> * **L249: “for a different values”, 250: Appendix B should be capitalized, 251: sufficiently frequently**:
> > Thank you. We have fixed it in the updated version.

---

> > ### Author Response · Authors · 2024-11-21
> >
> > * **the A100 GPU you are using is an unusual choice for this task**:
> > > : We agree with the reviewer that it is a good idea to test out the algorithm on different accelerators. Unfortunately, the cluster at our disposal only had A100 GPUs. Vanilla SGD also benefits equally from this hardware, so with think that the comparison remains fair.
> > * **Why did you choose 300 epochs for ImageNet? What happens with a lower (and more standard) number?**:
> > > We didn’t use a systematic approach to choose the number of epochs. After training ImageNet on ResNet18 for 160 epochs using sequential SGD and only obtaining ~62%, we went for a much higher number (actually 250, we have updated the PDF) of epochs when scaling to ResNet50. Due to the long training time, we didn’t do any parameter search and used the CIFAR100 parameters for the task, except the weight_decay which was lowered to 0.0035.
> > * **Please consider moving figures closer to where they are referenced.**:
> > > Thank you for the remark, we did our best and a new version has been uploaded.
> > * **Why no SGD baselines in tables?**:
> > Thank you for pointing this out. We have added an SGD baseline in the tables, for DDP comparison in Appendix
> > * **Your convergence curves in Figure 2 seem to suggest that in fact there is quite significant noise due to asynchrony (the staircase pattern). Can you show a similar curve for regular training?**:
> > > Sure, we have added some figures in the paper (Appendix A). Asynchrony certainly adds noise to training in general. To reduce that noise, we perform layer-wise updates, making the training more stable.
> > * **Can you maybe vary the degree of asynchrony?**:
> > >Async Block Updates (BU) and Async Layer-wise Updates (LU) exhibit different levels of staleness in their gradient updates. Async LU utilizes more recent gradients for updates, as it performs updates immediately after the gradients are computed at each layer. In contrast, Async BU updates the layers only when all the gradients for the respective layers are available. Our observations indicate that performing layer-wise updates yields the best trade-off between training time and accuracy.

---

> ### Comment · Reviewer_Mg9U · 2024-11-23
> **Rebuttal acknowledgment**
>
> Dear authors,
>
> Thank you for the detailed reply. I will update my score to "borderline below acceptance" to acknowledge your effort, but I think it is fairly clear that the paper is not ready for publication at this time.
>
> Best regards.

---

### Official Review · Reviewer_dUAZ · 2024-11-03

**Soundness:** 1
**Presentation:** 1
**Contribution:** 1
**Rating:** 3
**Confidence:** 4

**Summary:**

The authors propose to parallelises backpropagation updates across the layers of a model by asynchronously updating them
from multiple threads.

**Strengths:**

Various experimental settings are explored, including CIFAR, ImageNet and LSTM.

**Weaknesses:**

1. The authors fail to show the converging model performance in Table 1. It does not validate that the proposed method can achieve similar performance as the backpropagation baseline.
2. The authors fail to show the converging model performance of the backpropagation baseline in Table 2 as well.
3. No training curves are provided.
4. Limited novelty. Decoupled forward and backward is not new. Check pipeline parallelism [1,2]. In particular, [2] provided a convergence analysis for async pipeline.
5. Figure 1 clearly shows that the forward activation and backward gradient does not come from the same training data. Therefore it's intuitionally not sound to show that it could match the performance of backpropagation. Note that [2] makes sure they come from the same data to conduct the convergence analysis.
6. Convergence complexity not provided.

[1] Huang, Y., Cheng, Y., Bapna, A., Firat, O., Chen, D., Chen, M., Lee, H., Ngiam, J., Le, Q.V. and Wu, Y., 2019. Gpipe: Efficient training of giant neural networks using pipeline parallelism. Advances in neural information processing systems, 32.

[2] Xu, A., Huo, Z. and Huang, H., 2020. On the acceleration of deep learning model parallelism with staleness. In Proceedings of the IEEE/CVF Conference on Computer Vision and Pattern Recognition (pp. 2088-2097).

**Questions:**

1. Could you provide a clear illustration in Figure 2 like Gpipe to show how the forward and backward are pipelined?

---

> ### Author Response · Authors · 2024-11-21
>
> We thank the reviewer for feedback and suggestions for improvement. We respond to individual points below.
>
> * **The authors fail to show the converging model performance in Table 1. It does not validate that the proposed method can achieve similar performance as the backpropagation baseline.**
> > Thank you for the remark.
> We did compare with DDP, an implementation of Backpropagation for multiple devices, separately in the Appendix.
>
> * **The authors fail to show the converging model performance of the backpropagation baseline in Table 2 as well.**:
> > Thank you for the remark. we have added the accuracy of single-GPU SGD in the tables
>
> * **No training curves are provided.**:
> >Thanks for the suggestion, we have added some training curves in the appendix A
> * **Limited novelty. Decoupled forward and backward is not new. Check pipeline parallelism [1,2]. In particular, [2] provided a convergence analysis for async pipeline.**:
> >To the best of our knowledge, forward and backward passes have not yet been decoupled in data parallel settings. In contrast, attempts to achieve such decoupling in model or pipeline parallelism often result in challenges such as the recomputation of activations or the creation of bubbles, which can reduce computational efficiency. Moreover, our contribution goes beyond decoupling the forward and backward passes in data parallel settings. We introduce layer-wise updating, which is a premiere to the best of our knowledge.
> * **Figure 1 clearly shows that the forward activation and backward gradient do not come from the same training data. Therefore it's intuitionally not sound to show that it could match the performance of backpropagation. Note that [2] makes sure they come from the same data to conduct the convergence analysis.**:
> > In fully asynchronous data parallelism, the parameters' updates are usually done on stale gradients to make the training faster, our method is not different in that sense. Contrary to other algorithms, we perform layer-wise updates to reduce the staleness of the parameters and hence make the training more stable. Added training curves in Appendix A support these findings.
> * **Convergence complexity not provided.** :
> > We are uncertain about the meaning by convergence complexity here. However, we can confirm that the computational complexity of our method is the same as that of backpropagation (BP). Additionally, we have conducted a convergence analysis to validate our approach in section 5 and appendix B.

---

### Official Review · Reviewer_DHKJ · 2024-11-03

**Soundness:** 2
**Presentation:** 3
**Contribution:** 1
**Rating:** 3
**Confidence:** 4

**Summary:**

the papers proposes methodology to faster train NN using asynchronous updates
This is tackling the problem known in the literature as forward/backward lock
in particular the authors propose an asynchronous methodology for the training

**Strengths:**

The ideas are clearly explained.
The paper is easy to follow.
The paper covers an important (crucial from the  reviewer's humble opinion) topic with a high potential for impact.

**Weaknesses:**

This work presents a technique that is very close to techniques that have been known to the community for several years now (see https://proceedings.mlr.press/v70/jaderberg17a/jaderberg17a.pdf for instance).
This paper was followed by a long line of work on the topic.
This should be at least mentioned in the literature review.
Moreover, extensive numerical experiments comparing the proposed technique with SOTA asynchronous training techniques should support and highlight the strength/weaknesses of the algorithm. This is currently lacking in the paper.
Last, we would like to note that evaluating the techniques on resnet50 and lstm is not representative of the challenges faced when training modern ml systems (eg llms). The reviewer expects to see numerical results covering more of  such architectures.

**Questions:**

If the reviewer is not mistaken, the interactions between the threads in figure 1 would indeed lead to the creation of memory buffers.
The reviewers is curious whether the authors have investigated such "communication costs" of the technique.

---

> ### Author Response · Authors · 2024-11-21
>
> We thank the reviewer for feedback and suggestions for improvement. We respond to individual points below.
>
> * **This work presents a technique that is very close to techniques that have been known to the community for several years now (see https://proceedings.mlr.press/v70/jaderberg17a/jaderberg17a.pdf for instance). This paper was followed by a long line of work on the topic. This should be at least mentioned in the literature review.**:
> >Thank you for the literature suggestions. We referenced methods which followed model parallelism (Block local learning) from which pipeline parallelism is derived. However,
> We would like to highlight key differences between our method and Decoupled Neural Interfaces using Synthetic Gradients:
>    * Our method adheres to the data parallel paradigm, unlike model or pipeline parallelism. Specifically, we do not split the model into blocks distributed across devices; instead, we keep the entire model intact on each device
>    * We don't make use of any auxiliary network to get synthetic gradients
> * **Moreover, extensive numerical experiments comparing the proposed technique with SOTA asynchronous training techniques should support and highlight the strengths/weaknesses of the algorithm. This is currently lacking in the paper.**:
> > We conducted a comprehensive comparison of our approach with the most well-established methods that adhere to the data parallelism paradigm. Methods performing model and pipeline parallelism were left out for that reason.
> * **Last, we would like to note that evaluating the techniques on resnet50 and lstm is not representative of the challenges faced when training modern ml systems (eg llms). The reviewer expects to see numerical results covering more of such architectures.**:
> > Most current asynchronous data parallel methods compare on vision tasks, due to computational constraints. To train an LLM with 3x more GPUs is simply not feasible for us because we don’t have the computing resources available.
> Also, more common for LLMs is pipeline/tensor parallelism.
> This is a replacement for data parallelism (which is true for decentralised Hogwild! as well). Not too suited for LLMs which require pipeline/tensor parallelism due to their size.
> * **If the reviewer is not mistaken, the interactions between the threads in Figure 1 would indeed lead to the creation of memory buffers. The reviewer is curious whether the authors have investigated such "communication costs" of the technique.**:
> > No additional buffer is needed during the interactions between the threads for our approach. This is due to the application of layer-wise updates where the latest available parameters’ updates just locking-free override the current value of the parameter therefore no buffer is needed. This communication is further hidden due to parameters’ updates being interleaved with the backward computation.

---

> > ### Comment · Reviewer_DHKJ · 2024-11-26
> > **reply to the rebuttal**
> >
> > thank you for the precisions regarding the buffer.
> > Despite the (appreciated) replies to some of my concerns/comments, I still do not believe the contributions of this work deserve publication.
> > I will let my score at 3.

---

### Official Review · Reviewer_46Jx · 2024-11-06

**Soundness:** 2
**Presentation:** 2
**Contribution:** 2
**Rating:** 3
**Confidence:** 3

**Summary:**

Given the observation that a backward pass usually takes twice as long as a forward computation in the Backpropagation algorithm, this paper proposes to perform the backward and forward passes in parallel threads instead of sequentially. Two parallel threads perform backward passes concurrently, while a third continuously perform forward passes and asynchronously update the model. Given that 2 threads are dedicated to backward passes, the cost of performing those computations is hidden behind the forward passes and the training time for the model simply amount to the time to perform forward computations without stopping on a given number of minibatches.

**Strengths:**

* **Novel use of multi-threading to hide the backward computations:** to the best of my knowledge, the way multi-threading is used here to hide the backward computation time in a "modularized" version of the back-propagation algorithm is not common. However, it is unfortunately *not* the only way to leverage multi-threading to this end (see the Weaknesses).
* **Theoretical analysis of the method:** convergence guarantees are given by modeling the training dynamic using stochastic processes.

**Weaknesses:**

* **Incomplete literature review, lacking comparison with existing literature:** Update locking is a key subject in model parallel (pipeline parallel) as it leads to GPUs idling instead of computing, yet no mention nor comparison with pipeline parallel methods is done. In model parallel, one thread per module is used, each thread performs a repetition of *“receive activation from previous module, forward pass, send activation to next module, receive gradient from next module, backward pass and update local module, send gradient to previous module”* cycles. See e.g.,[[Qi et al., 2023]]( https://arxiv.org/pdf/2401.10241 ), [[Kosson et al., 2021]]( https://arxiv.org/pdf/2003.11666 ), [[Yang et al., 2020]]( https://arxiv.org/pdf/1910.05124 ), [[Chen et al., 2019]]( https://arxiv.org/pdf/1809.02839 ), [[Huang et al., 2019]]( https://arxiv.org/pdf/1811.06965 ).
* **Idea of performing two backward computations per forward is not new**: in the “zero bubble paper” the same justification as eq (1) is done to reduce the size of the bubble in pipeline parallelism (see Fig.2 and Fig.3 of [[Qi et al., 2023]]( https://arxiv.org/pdf/2401.10241 ) ).
* **No study of the memory requirement of the method is performed:** Where are the activation terms $y_m$ of equation (1) in the Algorithm 1? These are are highly important in the memory footprint of your method, given that storing activations is the reason why the memory footprint is the largest at the end of the forward pass (see e.g., [the pytorch blog article on GPU memory]( https://pytorch.org/blog/understanding-gpu-memory-1/ ) ) and that GPU memory is one of the main bottleneck nowadays to train large models.

* **Weak experimental results:** The accuracies reached on CIFAR10 for ResNet18 are not really state-of the art (they should be $>95\\%$), and experiments with pipeline parallel algorithms are lacking, as well as with Decoupled Parallel Backpropagation [[Huo et al., 2018]]( https://arxiv.org/pdf/1804.10574 ).

* **Writing sometimes unclear or imprecise**:
  * **line 117:** *"Decoupled Parallel Backpropagation does full Backpropagation but uses stale gradients in the blocks to avoid update locking."* Indeed, but so does your method, so where is the problem?
  * **Overly complicated analysis of speedup (Section 3.3)**: the analysis could be made simpler by saying that your method basically performs forward passes back to back, completely hiding backward computation in parallel threads, so the training time is roughly $T_2 = bT$. On the contrary, standard BP performs forward and backward passes back to back, so the training time is $T_1 = (1+\beta )b T$ and the speedup is equal to $1 + \beta$.

  * **Section 3.4 unclear:** I did not understand the meaning of the staleness. Are you referring to the length of the “pipeline step” (see, e.g. fig 2 of [[Kosson et al., 2021]]( https://arxiv.org/pdf/2003.11666 ) ), or the “bubble size” in pipeline parallel (see, e.g. fig 2. in  [[Huang et al., 2019]]( https://arxiv.org/pdf/1811.06965 ) ).
  * **typos:** *line 221*: “with with”, *line 194:* the second line of equation is incorrect.

**Questions:**

* In your experiments, how many data parallel replicas are used for Hogwild? And for yours?
* In your experiments, how does the memory usage compare between Hogwild and your method?

---

> ### Author Response · Authors · 2024-11-21
>
> We thank the reviewer for feedback and suggestions for improvement. We respond to individual points below.
>
> * **Incomplete literature review, lacking comparison with existing literature**:
> >Thank you for the literature suggestions. Our method falls under the category of (asynchronous) data parallelism rather than model parallelism. Unlike model parallel approaches, which split the model across devices, our approach processes data in parallel without distributing the model itself. As such, we allocated only a brief paragraph in the literature review to model parallelism.
> However, we acknowledge the importance of clarifying the distinction between our method and model parallel techniques, including those referenced (e.g., [Qi et al., 2023], [Kosson et al., 2021], [Yang et al., 2020], [Chen et al., 2019], [Huang et al., 2019]). To address this, we can add a concise discussion highlighting how our approach differs to model parallel frameworks.
>
> * **Idea of performing two backward computations per forward is not new**:
> >To our knowledge, this approach has not been applied to data parallelism before. Unlike the "zero bubble" method, which achieves near-zero idle time in pipeline parallelism (i.e. a form of model parallelism), our approach ensures that devices experience no idle time at all. We can add a comparison with this method in an updated version. Moreover, our contributions also heavily relies on the application of layer-wise updates as decoupling the forward and backward passes can be insufficient in itself.
>
> * **No study of the memory requirement of the method is performed**:
> >Similar to most data-parallel algorithms, our method does not require additional memory buffers to function. As with standard backpropagation, activations are stored during the forward pass for later usage in the backward pass for gradient computation. Our approach neither involves activation recomputation nor necessitates saving gradients, thereby avoiding additional memory overhead.
>
> * **Weak experimental results**:
> >We chose not to include comparisons with model or pipeline parallel algorithms, as they represent a different paradigm. Instead, our focus was on benchmarking our method against LPPSGD and LAPSGD, which are more appropriate for comparison since they also belong to the category of data parallel algorithms.
> For the 1-2% accuracy gap, we used the Libtorch library for our experiments to benefit from C++ multithreading capacities since true multithreading doesn’t yet exist in Python. We have noticed the gap between Python and C++ accuracies, most probably because the model implementation and data preparation steps were written by us. On CIFAR10, our baseline is the best accuracy reached by SGD: ~93% using Libtorch. We have added those values in the tables.
>
> * **Writing sometimes unclear or imprecise**:
>    * **line 117**:
>     > Thank you for pointing this out. We could have been more explicit in our explanation. Decoupled Parallel Backpropagation adopts a model/pipeline parallelism approach, making it somewhat orthogonal to our method. While our method, like any asynchronous data parallel algorithm, may use stale gradients, it does not split the model into blocks across devices. More importantly, our approach does not require additional buffers to store stale gradients.
> Unlike the approach in Jaderberg et al.'s paper, we do not add extra MLPs, modify the architecture, or use explicit approximations. Our method relies solely on stale gradients without introducing additional memory or architectural complexities.
>    * **Overly complicated analysis of speedup (Section 3.3)**:
>     > Thank you for the remark. We have made the analysis simpler as it should be while keeping key ideas and uploaded a new version.
>    * **Section 3.4 unclear**:
>     > By staleness, we mean the time difference between when a parameter update becomes available and when it is used during the next forward pass. We demonstrated that layer-wise updates can reduce parameter staleness by applying updates immediately as they become available. To the best of our knowledge, this approach has not been explored before and could potentially enhance the stability of methods like Hogwild!.
>    * **typos**: Thank you for pointing out these Errors. We have fixed them in the updated version.
> * **How many data parallel replicas are used for Hogwild? And for yours?**:
> >We have implemented our code in C++ using LibTorch (the C++ interface of the PyTorch library), so only one process is spawnedn but the different backward passes and performed in threads, 3 in this case. As stated in the paper, we spawned one process per GPU (over 3 GPUs) for both LPPSGD and LAPSGD to ensure a fair comparison with our method.

---

> > ### Author Response · Authors · 2024-11-21
> >
> > * **Memory usage**:
> > although no extra memory buffer is needed for our algorithm in theory, it uses in practice around 6-8% more memory than LAPSGD, LPPSGD  and sequential SGD depending on whether the gradient updates are broadcast to one GPU or all GPUs. The difference can also be explained by an implementation limitation since at the moment, we cannot safely deallocate the memory used by the gradients already used without deallocating other memories. We are exploring possible solutions for that.

---

> ### Comment · Reviewer_46Jx · 2024-11-27
>
> * **data vs model parallel:** Many "pipeline parallelism" methods are used in addition to data-parallelism to reach sufficiently large batch sizes when training. In this context, the "model parallelism" part could be implemented as standard parallel threads in a single machine, removing the physical split of model in separate GPUs to just several model splits in the same machine and thus falling back to straight data parallel. This has the advantage to introduce some level of asynchrony and parallelism in the forward/backward computations given that many algorithms in pipeline parallelism are designed for this purpose to maximize the GPU throughput. Given the rich literature in pipeline parallelism and that your method is designed for the same purpose, a very relevant baseline for your work would be to compare your method to all previous methods in model parallel, by using the same *algorithms* as previous methods but separate *threads* on a single machine for each "model split" in model parallel in the *implementation*.
>
> * **memory usage:** "*activations are stored during the forward pass for later usage in the backward pass for gradient computation."* this is indeed the case, but activations are also erased as soon as they are used in the backward pass to free memory space in standard BP. Given that your method uses two backward threads, this cannot be done as easily and trivially and would thus lead to a memory overhead. Explicitly writing which version of the model and activation are used in Algo.1 could help identify this problem.
>
> We thank the authors for their answer, but given that the main problems raised were not addressed, we decide to keep our score.

---

### Author Response · Authors · 2024-11-21

Dear Reviewers,

Thank you once again for your valuable comments and feedback. Your remarks have been extremely helpful in improving our work. We have taken time to address the concerns raised carefully and have posted detailed responses for your review along with an updated version of the paper.

We understand that this is a particularly busy period, and we truly appreciate any time you can spare to provide further feedback on whether our responses adequately address your concerns. If there are any additional comments or suggestions, we remain committed to addressing them promptly and thoroughly.

Thank you for your time and consideration.

Sincerely,
The Authors

---

### Meta-Review · Area_Chair_cG5S · 2024-12-20

**Metareview:**

The paper proposes an asynchronous method for neural network training that decouples forward and backward passes using multiple threads.
The main concerns that remained after the author feedback phase were on two key issues: First, overlap with existing pipeline parallelism literature that wasn't adequately addressed or compared against. Second, there were concerns about experimental rigor, including missing baselines against standard SGD, lower accuracy than standard benchmarks, and insufficient analysis of memory requirements and computational overhead.

While several of the discussions were moving in a positive direction in the feedback phase, unfortunately the main concerns couldn't be fully resolved yet.

We hope the detailed feedback helps to strengthen the paper for a future occasion.

**Additional Comments On Reviewer Discussion:**

The author feedback phase was useful as acknowledged by the reviewers. Some of the concerns however remained if the work is ready for the high bar of ICLR.

---

### Decision · Program_Chairs · 2025-01-22

Reject